# An H3K14ub-H3K9me3 feedback circuit governs heterochromatin spreading and inheritance in fission yeast

Takenori Toda[1,5], Junyao Zang[1,5], Hongyun Qi[2,5], Yimeng Fang[1,5], Peng Jiang[3,5], Chun-Min Shan [3], Jiemin Wong [4] & Songtao Jia [1]✉

Heterochromatin is a bistable chromatin state essential for genome stability and gene regulation. Its spreading and inheritance have long been explained by a "read-write" cycle in which histone methyltransferases bind pre-existing tri-methylation of histone H3 lysine 9 (H3K9me3) and propagate this mark to neighboring nucleosomes. However, the weak affinity and limited catalytic stimulation provided by H3K9me3 alone challenge this model. The fission yeast H3K9 methyltransferase Clr4 functions within the CLRC complex, which also catalyzes histone H3 lysine 14 ubiquitination (H3K14ub). Here we show that H3K14ub and H3K9me3 form a feedback loop: H3K14ub strongly stimulates Clr4 activity on nucleosomes, while both H3K14ub and H3K9me3 stabilize CLRC binding to chromatin. Even subtle perturbations that disrupt this feedback, such as mutating one of the three H3 genes to prevent ubiquitination or methylation, or impairing Clr3-mediated H3K14 deacetylation, compromises heterochromatin spreading and inheritance. Conversely, counteracting activities, such as H3K14 acetylation by Mst2 and H3K9 demethylation by Epe1, synergistically constrains heterochromatin expansion. Thus, rather than relying solely on the weak H3K9me3 "read-write" cycle, heterochromatin is maintained through an integrated circuit of ubiquitination, deacetylation, and methylation, which governs spreading and inheritance.

Heterochromatin is a transcriptionally repressive chromatin state that plays a critical role in maintaining genome stability by silencing repetitive elements, suppressing recombination, and regulating gene expression programs[1–3]. Dysregulation of heterochromatin has been linked to a wide range of human diseases, while its targeted modulation can enhance cancer immunotherapy by reactivating repeat expression and promoting immune recognition of tumor cells[4,5].

Heterochromatin is characterized by specific chromatin features, including hypoacetylated histones and methylation of histone H3 on lysine 9. Its assembly involves three interconnected phases: initiation, spreading, and inheritance[1,3,6]. Initiation occurs at defined nucleation sites, where sequence-specific factors or non-coding RNAs recruit histone-modifying enzymes. Once established, heterochromatin spreads into adjacent regions independent of DNA sequence, resulting in extended repressive domains. These domains can then be stably

[1]Department of Biological Sciences, Columbia University, New York, NY, USA. [2]State Key Laboratory of Molecular Biology, Shanghai Key laboratory of Molecular Andrology, Institute of Biochemistry and Cell Biology, Shanghai Institutes for Biological Sciences, Chinese Academy of Sciences, Shanghai, China. [3]Department of Agri-microbiomics and Biotechnology, State Key Laboratory of Microbial Diversity and Innovative Utilization, Institute of Microbiology, Chinese Academy of Sciences, Beijing, China. [4]Shanghai Key Laboratory of Regulatory Biology, Fengxian District Central Hospital-ECNU Joint Center of Translational Medicine, Institute of Biomedical Sciences, School of Life Sciences, East China Normal University, 500 Dongchuan Road, Shanghai, China. [5]These authors contributed equally: Takenori Toda, Junyao Zang, Hongyun Qi, Yimeng Fang, and Peng Jiang. ✉e-mail: songtao.jia@columbia.edu

inherited across cell divisions, even in the absence of the initiating signal. While heterochromatin initiation has been well characterized, the molecular mechanisms underlying its spreading and inheritance remain incompletely understood.

A defining feature of heterochromatin is its bistable behavior, in which genomic regions can switch between active and silent states yet maintain sharp boundaries between them[7,8]. Bistability usually arises from the interplay of reinforcing and antagonizing activities that generate switch-like behavior[9]. However, the precise molecular circuits that confer bistability to heterochromatin domains remain poorly defined.

The fission yeast *Schizosaccharomyces pombe* has been widely used to dissect the molecular mechanisms of heterochromatin assembly. In this organism, heterochromatin forms primarily at pericentric repeats, telomeres, and the silent mating-type region, all of which share common repetitive elements[10]. At these sites, RNA interference (RNAi) machinery processes repeat-derived transcripts into small interfering RNAs (siRNAs), which guide the H3K9 methyltransferase Clr4 to initiate heterochromatin formation[11-13] (Fig. S1). Heterochromatin then spreads from initiation sites through a "read-write" mechanism in which Clr4 uses its chromodomain to bind existing H3K9me3 and methylates adjacent nucleosomes[14,15]. The "read-write" coupling is also critical for heterochromatin inheritance: during DNA replication, parental histones marked with H3K9me3 are distributed to daughter strands[16,17], where they guide Clr4 to restore the methylation pattern on newly synthesized histones[15,18,19] (Fig. S1). However, the binding of Clr4 chromodomain to H3K9me3 and the stimulation of Clr4 activity by H3K9me3 are relatively weak[14,15], and whether they form the sole basis of heterochromatin bistability is unknown.

Central to heterochromatin formation is the Clr4 histone H3K9 methyltransferase[20,21]. It also functions as part of the CLRC E3 ubiquitin ligase complex, which includes Clr4, Cul4, Rik1, Raf1, and Raf2[22-26]. Like other Cullin-based E3 complexes, Cul4 forms the scaffold, Raf1 is the presumptive substrate recognition subunit, and Rik1 links Raf1 to Cul4[27,28]. The function of Raf2 is currently unknown, but it has been implicated to regulate CLRC integrity[29]. In addition to its methyltransferase activity, CLRC catalyzes monoubiquitylation of histone H3 on lysine 14 (H3K14ub), a modification that enhances Clr4 activity in vitro[30-33]. However, the genomic distribution and biological function of H3K14ub in vivo have remained unclear due to the lack of a specific detection reagent.

Heterochromatin spreading and inheritance also require the histone deacetylase Clr3, which specifically removes acetyl groups from H3K14[34-36]. Clr3-mediated deacetylation has been proposed to stabilize heterochromatin by reducing histone turnover[37-39], but it remains unknown whether this activity also promotes heterochromatin through regulation of H3K14ub, which is mutually exclusive with H3K14 acetylation.

In this study, we define a chromatin-based positive feedback loop between H3K14 ubiquitylation and H3K9 methylation that is essential for heterochromatin spreading and inheritance. Using a H3K14ub-specific antibody[40], we show that H3K14ub co-localizes with H3K9me3 across the genome and depends on CLRC and Clr4 enzymatic activity for deposition. Conversely, H3K14ub promotes H3K9me3 on nucleosomes via stimulation of Clr4, establishing a mutually reinforcing circuit. This loop is modulated by Clr3-mediated H3K14 deacetylation, which controls the extent of spreading and inheritance. In addition, Raf1 overexpression can enhance feedback strength and restore heterochromatin spreading and inheritance when the circuit is compromised. Finally, we show that H3K9 demethylase Epe1 and H3K14 acetyltransferase Mst2 function together to limit feedback activation and prevent unchecked heterochromatin expansion. The balance between reinforcing and antagonizing activities thus establishes a bistable system. These findings reveal a previously unrecognized layer of crosstalk between histone deacetylation, ubiquitylation, and methylation that promotes robust heterochromatin formation and maintenance.

## Results

### H3K14ub stimulates Clr4 enzymatic activity on nucleosome substrates in vitro

Previous studies investigating the effects of H3K14ub on Clr4 enzymatic activity in vitro primarily used histone tail peptides as substrates[30-33]. However, whether H3K14ub modulates Clr4 activity in different substrate contexts has not been compared. To address this question, we purified recombinant Clr4 SET domain (residues 190-490) as well as a mutant version containing mutations in three phenylalanine residues (F256A, F310A, and F427A, hereafter referred to as Clr4-3FA) known to disrupt the interaction between Clr4 and H3K14ub[31] (Fig. 1a, b). We performed an in vitro histone methyltransferase assay using $^{3}$[H]-S-adenosyl-methionine (SAM) and recombinant substrates, including H3/H4 tetramers, wild-type mononucleosomes, and H3K14ub-modified mononucleosomes. Clr4 exhibits robust methyltransferase activity toward H3/H4 tetramers but shows minimal activity towards unmodified mononucleosomes under the conditions tested, suggesting that nucleosome DNA inhibits Clr4 function (Fig. 1c). In contrast, Clr4 displays strong activity towards H3K14ub mononucleosomes, indicating that H3K14ub alleviates the inhibitory effects of nucleosome DNA.

Further in vitro analyses reveal that the Clr4-3FA mutant retains activity on H3/H4 tetramers but fails to respond to stimulation by H3K14ub mononucleosomes (Fig. 1c). These results suggest that the interaction between Clr4 and H3K14ub is critical for stimulating Clr4 enzymatic activity in the context of nucleosomes. Notably, this stimulatory effect is specific to H3K14ub, as Clr4 shows little activity toward recombinant H3K18ub mononucleosomes or H2BK120ub mononucleosomes (Fig. 1d).

### H3K14ub is enriched at heterochromatin in vivo

Biochemical studies have shown that the CLRC complex catalyzes H3K14ub (Fig. 2a)[33]. However, the genomic distribution of H3K14ub in vivo has not been characterized. To address this question, we generated a highly specific antibody against H3K14ub[40]. Western blot analysis using recombinant nucleosomes with ubiquitin at different lysine residues confirmed the antibody's specificity: it robustly recognizes H3K14ub but does not cross-react with H3K18ub or H2BK120ub (Fig. 2b).

Chromatin immunoprecipitation followed by sequencing (ChIP-seq) reveals that H3K14ub is highly enriched at heterochromatin domains, including pericentric repeats, telomeres, and the silent mating-type region, with a distribution closely resembling that of H3K9me3 (Fig. 2c). Both marks are undetectable in *rik1Δ* cells (Fig. 2c), indicating a strict dependence on CLRC for their deposition. Consistent with this, ChIP-qPCR confirms that H3K14ub is enriched at pericentric *dh* repeats, but is abolished in strains carrying the *K14R* substitution in all histone H3, as well as in CLRC mutants (*raf1Δ*, *raf2Δ*, and *rik1Δ*) (Fig. 2d, e). All of these mutants also show complete loss of H3K9me3 (Fig. 2d, e). These findings establish the genome-wide map of H3K14ub in fission yeast and demonstrate that CLRC is indispensable for its deposition in vivo, consistent with a recent study[41]. The striking co-localization of H3K14ub with H3K9me3 further suggests that the two modifications act in concert to define heterochromatin domains, a feature also conserved in mammals[40]. The results are consistent with previous mass spectrometry analyses demonstrating the coexistence of H3K14ub and H3K9me3[33], and extend those findings by providing genome-wide localization at high resolution.

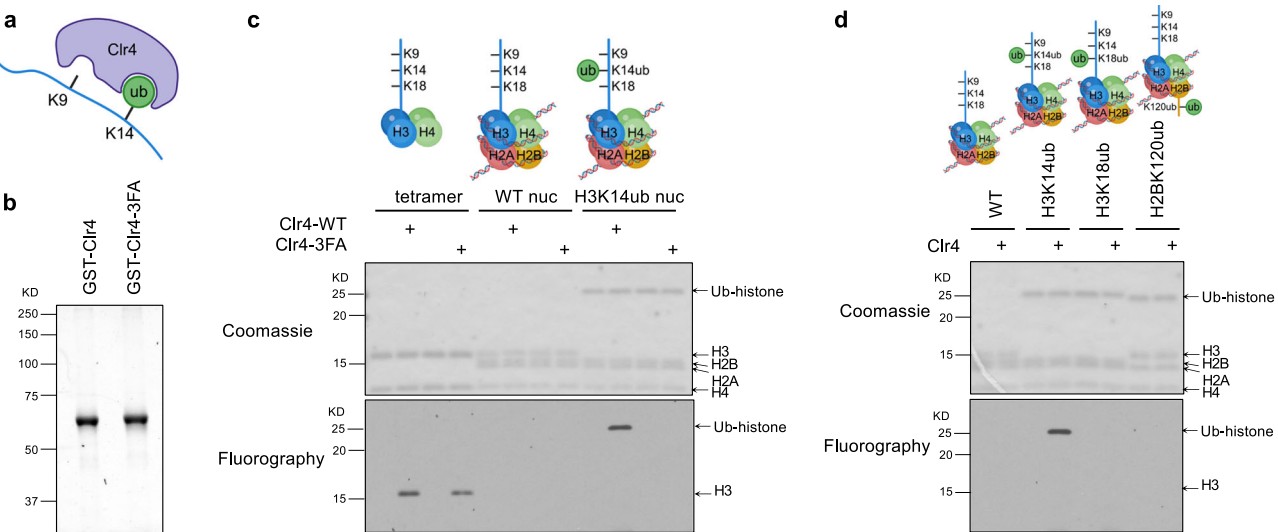

**Fig. 1 | H3K14ub stimulates Clr4 enzymatic activity on nucleosome substrates.**
**a** Schematic diagram of Clr4 and the H3 tail. Created in BioRender. Jia, S. (2026)
https://BioRender.com/x7jvxqt. **b** SDS-PAGE analysis of Clr4 protein and mutants
(in the context of GST-Clr4 190-490). The gel is stained with Coomassie blue. The
experiment was performed once. **c**, **d** In vitro histone methyltransferase assays
using Clr4-SET, ³H-SAM, recombinant H3/H4 tetramers, or indicated recombinant

mononucleosomes. The reaction product is resolved by SDS-PAGE, stained with
Coomassie blue (top), and processed for fluorography (bottom). The experiments
were performed twice with similar results. Diagram created in BioRender. Jia, S.
(2026) https://BioRender.com/x7jvxqt. Source data are provided as a Source
Data file.

## H3K9me is required for H3K14ub in vivo

To further dissect the relationship between these two modifications,
we also examined whether H3K9 methylation is required for H3K14ub.
ChIP analyses reveal that both H3K14ub and H3K9me3 are lost from
pericentric *dh* repeats in *clr4Δ* cells (Fig. 3a, b), as well as in the cata-
lytically inactive mutant *clr4-Y451N* mutant, which abolishes H3K9
methylation without affecting CLRC integrity[42] (Fig. 3a, b). Similarly,
the *H3K9R* substitution in all histone H3 genes abolishes H3K14ub,
demonstrating that H3K9me is essential for H3K14 ubiquitylation
in vivo (Fig. 3a, b).

Mechanistically, H3K9me3 stabilizes CLRC recruitment to
chromatin[15]. Indeed, ChIP analysis shows that Rik1, a core subunit of
CLRC, is no longer enriched at *dh* repeats in *clr4Δ*, *clr4-Y451N*, or *H3K9R*
cells (Fig. 3c). Two distinct mechanisms mediate Clr4 recognition of
modified nucleosomes: its chromodomain binds H3K9me3, an inter-
action disrupted by the W31G mutation[15], and its SET domain binds
H3K14ub, an interaction abolished by the 3FA mutation[31,32] (Fig. 3d).
ChIP analyses show that both *clr4-W31G* and *clr4-3FA* mutations
markedly reduce H3K9me3, H3K14ub, and Rik1 levels at *dh* repeats
(Fig. 3e–g). In addition, ChIP analyses show that Clr4 localization to *dh*
repeats is abolished in *H3K9R* and *H3K14R* cells (Fig. S2a), and co-
immunoprecipitation analysis shows that the Clr4-Rik1 interaction is
not affected by *H3K9R* or *H3K14R* substitutions (Fig. S2b). These results
demonstrate that recruitment of the entire CLRC complex is depen-
dent on both H3K9me and H3K14ub.

Together, these results uncover a positive feedback loop between
H3K14ub and H3K9me3: H3K14ub stimulates Clr4 activity to promote
H3K9me3, while both H3K9me3 and H3K14ub enhance CLRC recruit-
ment to heterochromatin, thereby facilitating further H3K14 ubiqui-
tylation and H3K9 methylation (Fig. 3h).

## H3K14ub-H3K9me3 feedback is critical for heterochromatin spreading and inheritance

Heterochromatin assembly proceeds through initiation, spreading,
and inheritance (Fig. S1)[1,3]. Spreading and inheritance rely on pre-
existing modifications, and therefore, are expected to be particu-
larly sensitive to perturbations of the H3K14ub-H3K9me3
feedback loop.

Fission yeast encodes three histone H3 genes (*hht1⁺*, *htt2⁺*, and
*hht3⁺*) expressed at similar levels[42,43] (Fig. 4a). If this feedback loop is
critical, then single-copy mutations that prevent H3 ubiquitination
(H3K14R) or methylation (H3K9R) should impair spreading or inheri-
tance, even in the presence of two wild-type copies (Fig. 4b). To test
this, we analyzed *hht3-K9R* and *hht1-K14R* mutants, each tagged with
C-terminal Flag at their endogenous loci (Fig. 4a). Western blot con-
firms that mutant H3 proteins are expressed at levels comparable to
their wild type counterparts (Fig. 4c).

To monitor heterochromatin function, we used *ade6⁺* reporter
genes inserted at distinct chromosomal locations. Reporter silencing
yields red colonies on low adenine (YE) medium, whereas expression
results in white or pink colonies. At the *otr::ade6⁺* reporter within
pericentric repeats, where heterochromatin is initiated by the RNA
interference (RNAi) pathway[11,12], single-copy H3 mutations (*hht3-K9R*
and *hht1-K14R*) cause only modest silencing defects, as indicated by
mostly red colonies (Fig. 4d) and slight reductions in H3K14ub and
H3K9me3 (Fig. 4e). These results suggest that initiation sites are rela-
tively resistant to mild perturbations of the feedback loop.

In contrast, the *mat3::ade6⁺* reporter, located in the silent mating
type region but distant from the *cenH* initiation element, specifically
measures heterochromatin spreading. Both single-copy H3 mutations
show strong spreading defects (white colonies, Fig. 4f), accompanied
by loss of H3K14ub and H3K9me3 at the reporter (Fig. 4g).

The *KΔ::ade6⁺* reporter, in which *cenH* is replaced with *ade6⁺*,
specifically measures heterochromatin inheritance (Fig. 4h). In this
context, silencing depends exclusively on parental histone-mediated
inheritance[44]. Both single-copy H3 mutations cause a strong loss of
silencing (Fig. 4h), accompanied by loss of H3K14ub and H3K9me3 at
the reporter as well (Fig. 4i).

These results suggest that local saturation of both marks is
necessary to sustain read-write coupling, and that interference with
even one third of total H3 can perturb this balance[45,46].

## The acetylation status of H3K14 regulates H3K14ub-H3K9me3 feedback

The H3K14ub-H3K9me3 feedback loop relies on the availability of
unmodified H3K14, since acetylation at this residue blocks

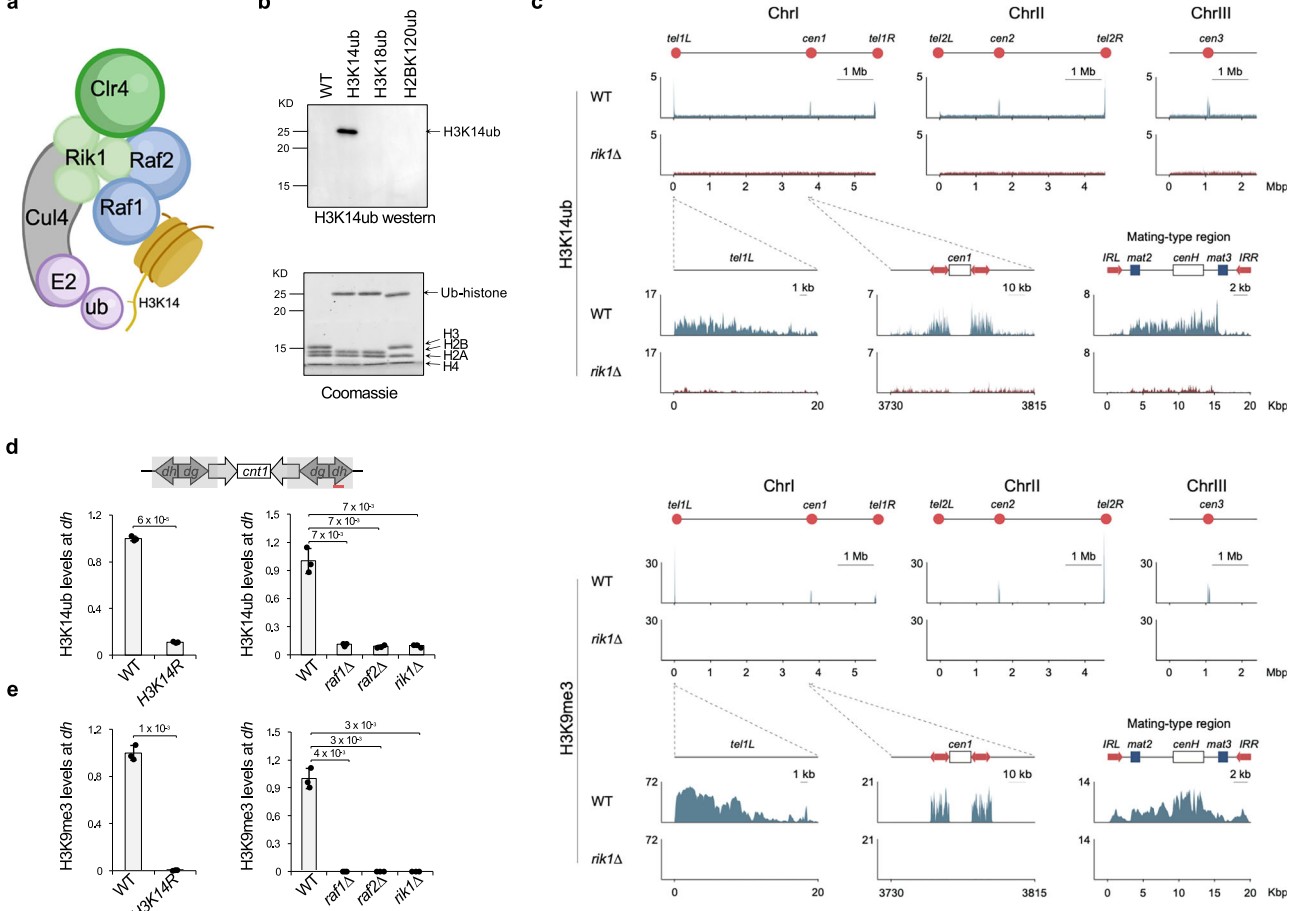

**Fig. 2 | CLRC regulates H3K14ub in vivo. a** Schematic diagram of CLRC. Created in BioRender. Jia, S. (2026) https://BioRender.com/ibuojdu. **b** Top, western blot analysis of recombinant nucleosomes with mono-ubiquitination at different positions with a H3K14ub antibody. Bottom, the recombinant nucleosomes were resolved by SDS-PAGE and stained with Coomassie blue. The experiment was performed once. **c** ChIP-seq analysis of H3K14ub and H3K9me3 across the fission yeast genome. The x-axis indicates genomic coordinates, and the y-axis indicates normalized read density (reads per million mapped reads). Schematic diagrams of the three chromosomes are shown on top, with key features (centromeres and telomeres) indicated by red circles. Regions surrounding the left telomere of chromosome I (*tel1L*), centromere I (*cen1*), and the silent mating-type region are enlarged and shown at the bottom. Note that the silent mating-type region is not represented in the chromosome II assembly. **d** ChIP analyses of H3K14ub levels at pericentric *dh* repeats. Data are presented as mean ± s.d. from 3 biological replicates. Statistical significance was assessed using two-tailed unpaired Student's *t*-tests for the indicated pairwise comparisons. Exact *P*-values are indicated above each comparison. **e** ChIP analyses of H3K9me3 levels at pericentric *dh* repeats. Data are presented as mean ± s.d. from 3 biological replicates. Statistical significance was assessed using two-tailed unpaired Student's *t*-tests for the indicated pairwise comparisons. Exact *P*-values are indicated above each comparison. Source data are provided as a Source Data file.

ubiquitination. We therefore investigated the role of Clr3, a histone H3K14deacetylase[34–36], in regulating this process. ChIP analysis reveals that *clr3Δ* causes only modest reductions in H3K14ub and H3K9me3 levels at pericentric *dh* repeats (Fig. S3a, b). It also results in enhanced enrichment of Rik1 and Clr4 at *dh* repeats (Fig. S3c, d).

Clr3-mediated H3K14 deacetylation is particularly important for heterochromatin spreading at the silent mating-type locus (Fig. 5a)[36]. In *clr3Δ* cells, both H3K14ub and H3K9me3 are modestly reduced at *cenH* but completely lost at *mat2P* (Fig. 5a, b), indicating that H3K14ub spreads together with H3K9me3. Both Rik1 and Clr4 show strong enrichment at *cenH*, but are absent at *mat2P* (Figs. 5c, S3e, f).

Previous studies have shown that *clr3Δ* increases transcription of repetitive DNA, thereby enhancing RNAi-dependent recruitment of CLRC to heterochromatin initiation sites[15,47], consistent with our findings. However, despite this increased recruitment at nucleation sites, CLRC fails to spread to distal regions such as *mat2P*. This uncoupling of initiation from spreading demonstrates that RNAi-mediated recruitment alone is insufficient to sustain heterochromatin propagation, and

that effective spreading of CLRC requires an intact H3K14ub-H3K9me3 feedback loop.

The modest reduction of H3K14ub and enhanced enrichment of CLRC at *dh* and *cenH* in *clr3Δ* cells suggest that elevated CLRC levels may partially compensate for the loss of deacetylation by outcompeting histone acetyltransferases for H3K14 (Fig. 5d). To test this idea, we overexpressed Raf1, the presumptive substrate recognition subunit of CLRC, by placing a *Flag-raf1*[+] construct under the control of the *ade6* promoter (*OE-raf1*[+]) (Fig. 5e).

In *clr3Δ* cells, the *mat3::ade6*[+] reporter is expressed, giving rise to white colonies on YE medium. In contrast, *OE-raf1*[+] *clr3Δ* cells form pink colonies (Fig. 5f), and partially restored H3K14ub and H3K9me3 at *mat3::ade6*[+] (Fig. 5g). Strikingly, *OE-raf1*[+] also partially rescued the spreading defects of *hht3-K9R* or *hht1-K14R*, as shown by pink colonies on YE medium (Fig. 5f) and restoration of H3K14ub and H3K9me3 at the reporter (Fig. 5g). In contrast, *OE-raf1*[+] does not rescue silencing defects of *swi6Δ*, consistent with Swi6 functions downstream of H3K9me3 (Fig. 5f). It also cannot rescue silencing defects of *clr4-W31G*

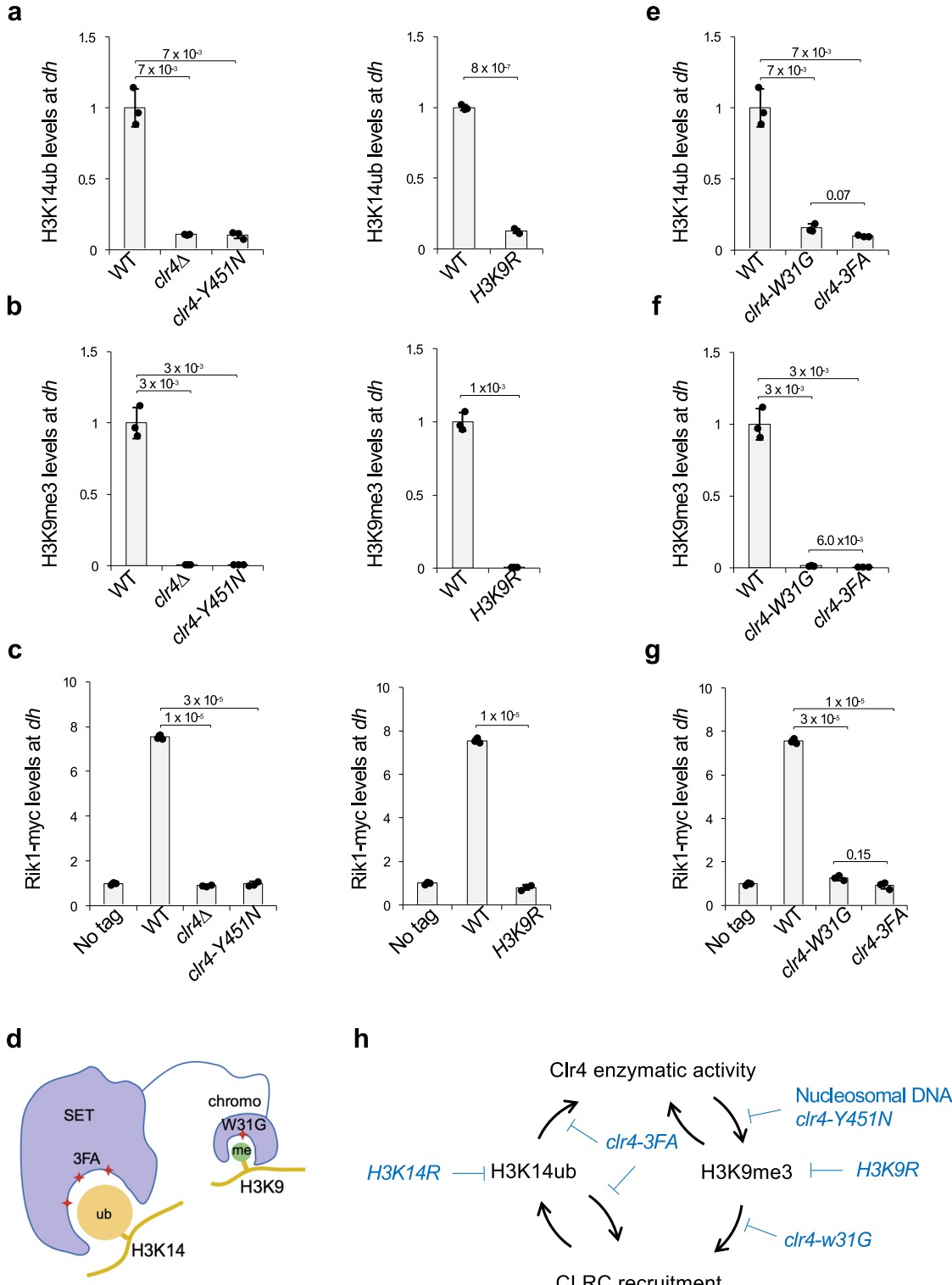

**Fig. 3 | H3K14ub in vivo requires Clr4-mediated H3K9 methylation. a, e** ChIP analyses of H3K14ub levels at pericentric *dh* repeats. Data are presented as mean ± s.d. from 3 biological replicates. Statistical significance was assessed using two-tailed unpaired Student's *t*-tests for the indicated pairwise comparisons. Exact *P*- values are indicated above each comparison. **b, f** ChIP analyses of H3K9me3 levels at pericentric *dh* repeats. Data are presented as mean ± s.d. from 3 biological replicates. Statistical significance was assessed using two-tailed unpaired Student's *t*-tests for the indicated pairwise comparisons. Exact *P*- values are indicated above each comparison. **c, g** ChIP analyses of Rik1-myc levels at pericentric *dh* repeats.

Data are presented as mean ± s.d. from 3 biological replicates. Statistical significance was assessed using two-tailed unpaired Student's *t*-tests for the indicated pairwise comparisons. Exact *P*- values are indicated above each comparison. **d** Schematic diagram of the Clr4 chromodomain and SET domain and their interaction with H3K9me3 and H3K14ub, respectively. Created in BioRender. Jia, S. (2026) https://BioRender.com/ibuojdu. **h** Schematic diagram of the H3K14ub-H3K9me3 feedback loop, highlighting a stronger contribution by H3K14ub. Source data are provided as a Source Data file.

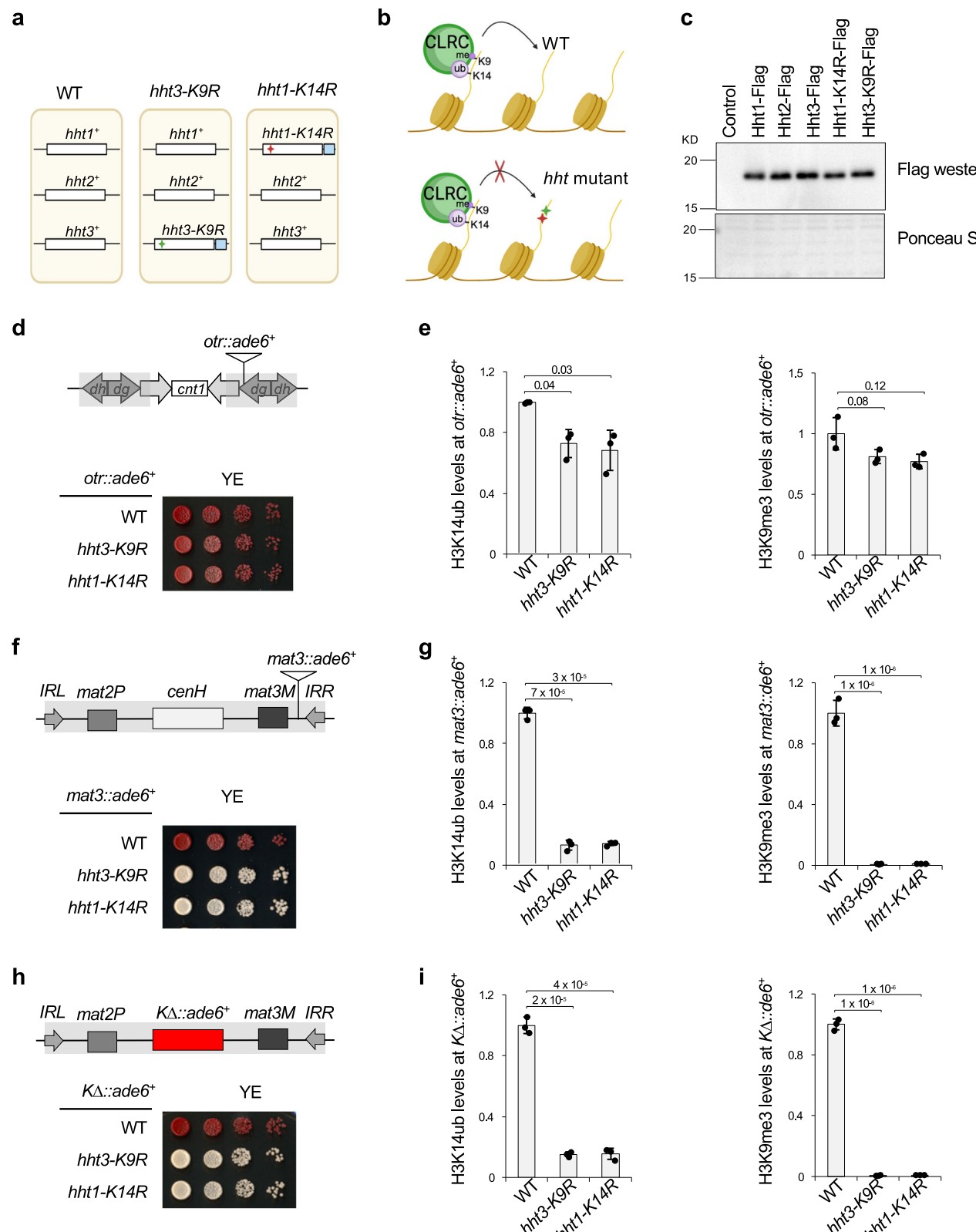

**Fig. 4 | The H3K14ub-H3K9me3 feedback loop is critical for heterochromatin spreading and inheritance. a** Schematic diagram of histone H3 genes and mutations used. Red and green stars indicate *K14R* and *K9R* mutations, respectively, and blue boxes indicate the Flag tag. Created in BioRender. Jia, S. (2026) https://BioRender.com/9h8l1be. **b** Schematic diagram of how single-copy H3 mutations block heterochromatin spreading and inheritance. Created in BioRender. Jia, S. (2026) https://BioRender.com/9h8l1be. **c** Top: Western blot of fission yeast cell extracts with a Flag antibody. Bottom, Ponceau S stain of the membrane. The experiment was performed once. **d**, **f**, **h** Serial dilution analysis of indicated strains to measure the expression of reporter genes. **e**, **g**, **i** ChIP analyses of H3K14ub and H3K9me3 levels at the reporter. Data are presented as mean ± s.d. from 3 biological replicates. Statistical significance was assessed using two-tailed unpaired Student's *t*-tests for the indicated pairwise comparisons. Exact *P*-values are indicated above each comparison. Source data are provided as a Source Data file.

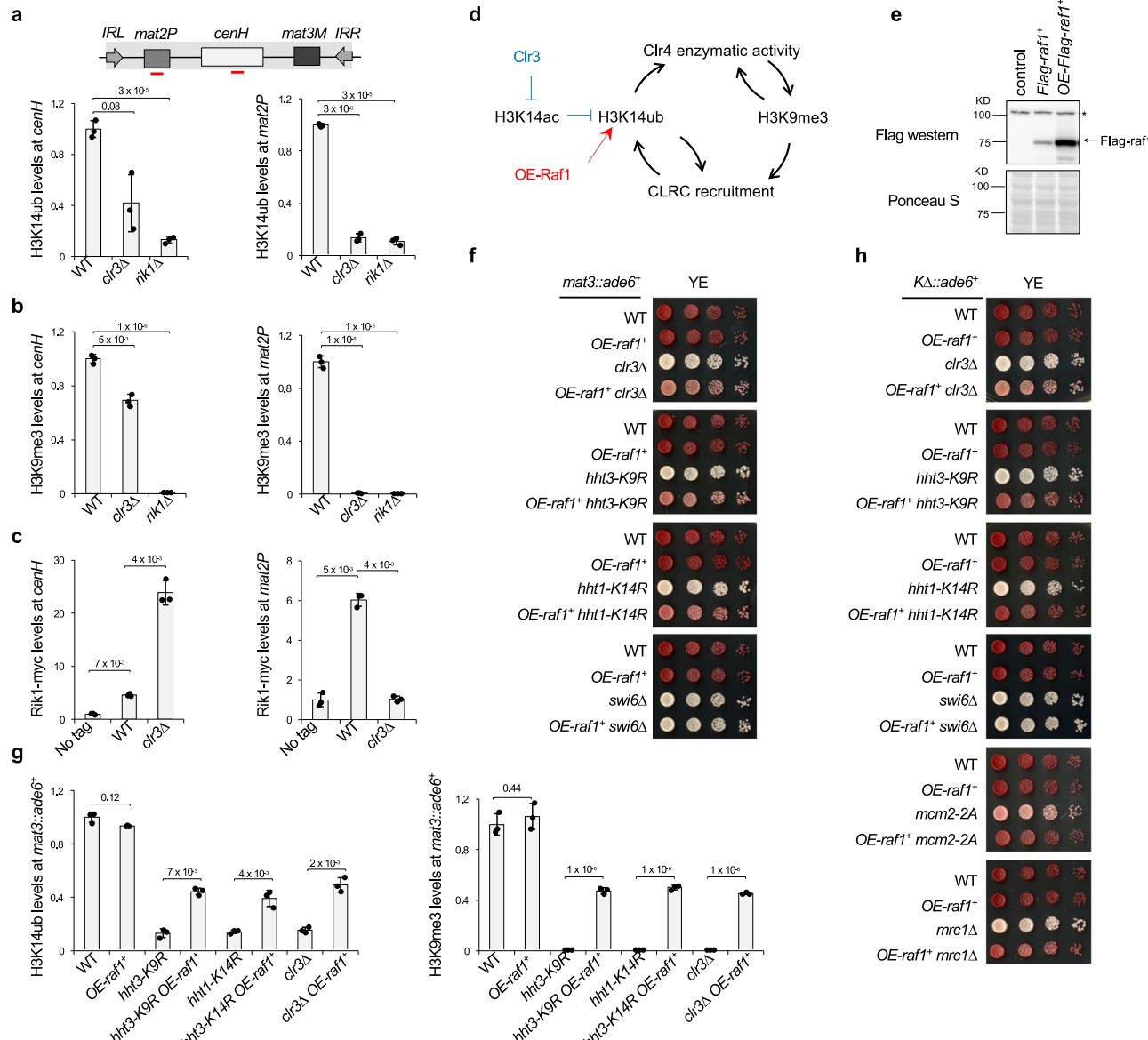

**Fig. 5 | Clr3 regulates H3K14ub-H3K9me3 feedback loop and heterochromatin spreading/inheritance. a** Top, schematic diagram of the silent mating-type region. Red bars indicate the position of PCR fragments used in qPCR. Bottom, ChIP analyses of H3K14ub levels at *cenH* and *mat2P*. Data are presented as mean ± s.d. from 3 biological replicates. Statistical significance was assessed using two-tailed unpaired Student's *t*-tests for the indicated pairwise comparisons. Exact *P*- values are indicated above each comparison. **b** ChIP analyses of H3K9me3 levels at *cenH* and *mat2P*. Data are presented as mean ± s.d. from 3 biological replicates. Statistical significance was assessed using two-tailed unpaired Student's *t*-tests for the indicated pairwise comparisons. Exact *P*-values are indicated above each comparison. **c** ChIP analyses of Rik1-myc levels at *cenH* and *mat2P*. Data are presented as mean ± s.d. from 3 biological replicates. Statistical significance was assessed using two-tailed unpaired Student's *t*-tests for the indicated pairwise comparisons. Exact

*P*-values are indicated above each comparison. **d** Schematic diagram of the H3K14ub-H3K9me3 feedback loop, highlighting its regulation by H3K14ac, Clr3 and OE-Raf1. **e** Top, Top: western blot of fission yeast cell extract with a Flag antibody. * represents a non-specific band that also serves as a loading control. Bottom, Ponceau S stain of the membrane. The experiment was performed twice with similar results. **f** Serial dilution analysis of indicated strains to measure the expression of *mat3::ade6+*. **g** ChIP analysis of H3K14ub and H3K9me3 levels at *mat3::ade6+*. Data are presented as mean ± s.d. from 3 biological replicates. Statistical significance was assessed using two-tailed unpaired Student's *t*-tests for the indicated pairwise comparisons. Exact *P*-values are indicated above each comparison. **h** Serial dilution analysis of indicated strains to measure the expression of *KΔ::ade6+*. Source data are provided as a Source Data file.

or *clr4-3FA*, suggesting that *OE-raf1+* exerts its effects through the H3K14ub-H3K9me3 feedback loop (Fig. S3g, h).

Raf1 overexpression also partially restored heterochromatin inheritance defects in *clr3Δ*, *hht3-K9R* or *hht1-K14R* mutants, as well as in *mcm2-2A* or *mrc1Δ*, which disrupt proper parental histone segregation to the lagging strand during DNA replication[45,48–50] (Fig. 5h). These results establish that H3K14ac modulates the strength of the H3K14ub-H3K9me3 feedback loop to support both spreading and inheritance.

## Regulation of the H3K14ub-h3K9me3 feedback loop by Mst2 and Epe1

If positive feedback drives heterochromatin spreading and inheritance, then negative regulatory mechanisms are equally important to prevent uncontrolled heterochromatin expansion. Two major heterochromatin antagonists are the histone acetyltransferase Mst2, which is specific for H3K14, and the JmjC-domain protein Epe1, a putative H3K9 demethylase[18,19,51–56]. Although their biochemical activities differ, the H3K14ub-H3K9me3 feedback loop explains their shared

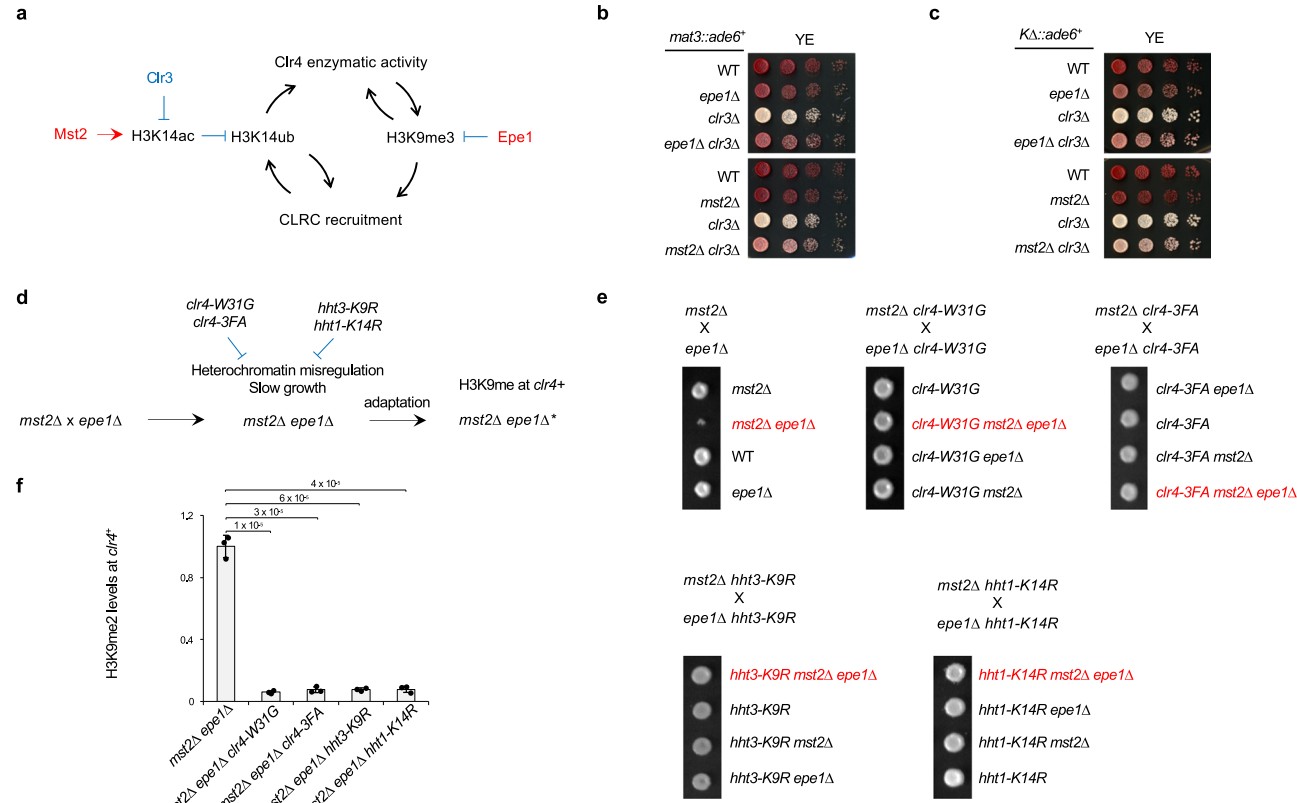

**Fig. 6 | Negative regulation of the H3K14ub and H3K9me3 is critical for heterochromatin stability. a** Schematic diagram of the H3K14ub-H3K9me3 feedback loop, highlighting its regulation by Mst2 and Epe1. **b** Serial dilution analysis of indicated strains to measure the expression of *mat3::ade6+*. **c** Serial dilution analysis of indicated strains to measure the expression of *KΔ::ade6+*. **d** Schematic diagram of the role of the effects of *mst2Δ epe1Δ* in regulating heterochromatin. **e** Top, strains used for genetic crosses. Bottom, the growth of four progenies from one tetrad. For

simplicity and better comparison, only one tetrad of tetratype was shown. Colonies containing both *mst2Δ* and *epe1Δ* are labeled red. **f** ChIP analysis of H3K9me2 levels at *clr4+*. Data are presented as mean ± s.d. from 3 biological replicates. Statistical significance was assessed using two-tailed unpaired Student's *t*-tests for the indicated pairwise comparisons. Exact *P*-values are indicated above each comparison. Source data are provided as a Source Data file.

phenotypes: Mst2-mediated H3K14ac counteracts Clr3-mediated H3K14 deacetylation, therefore inhibiting H3K14ub, while Epe1 directly erases H3K9me3 (Fig. 6a). Consistent with this logic, either *mst2Δ* or *epe1Δ* suppresses the silencing defects of *clr3Δ* at both *mat3::ade6+* and *KΔ::ade6+* (Fig. 6b, c), indicating that enhancing one arm of the loop can compensate for weakening another.

Loss of both Mst2 and Epe1, however, removes both inhibitory constraints, causing excessive heterochromatin spreading that silences essential genes and slows growth[51,57] (Fig. 6d, e). This stress triggers a compensatory response in which H3K9me accumulates at the *clr4+* locus, down regulating Clr4 expression[51,57] and allowing cells to recover from heterochromatic stress (Fig. 6f). Our model predicts that simultaneous loss of Mst2 and Epe1 synergistically activates the H3K14ub-H3K9me3 feedback loop, driving heterochromatin expansion and triggering this adaptive response.

To test this hypothesis, we examined H3K14ub and H3K9me3 levels at a region outside the boundary element of centromere I (*IRC1R*)[58]. Because *mst2Δ epe1Δ* rapidly generate epigenetically silenced *clr4+*, we analyzed *mst2Δ epe1Δ swi6Δ* cells, which tolerate elevated H3K9 methylation without triggering the formation of the epigenetic suppressor[51]. ChIP analyses show that *mst2Δ* or *epe1Δ* alone has no effects on H3K14ub or H3K9me3 outside of *IRC1R* (Fig. S4a), because the *IRC1R* boundary effectively blocks heterochromatin spreading. In contrast, in *mst2Δ epe1Δ swi6Δ* cells, both H3K14ub and H3K9me3 levels dramatically increased outside *IRC1R*, suggesting that supercharging the H3K14ub-H3K9me3 feedback loop overrides the heterochromatin boundary (Fig. S4a).

Either *mst2Δ* or *epe1Δ* results in an increase in H3K9me at facultative heterochromatin islands, such as *mei4+*[51,59]. Although we did not detect a significant increase in H3K14ub and only a modest increase in H3K9me3 in the single mutants, we observed a strong increase in both H3K14ub and H3K9me3 at *mei4+* in *mst2Δ epe1Δ swi6Δ* cells (Fig. S4b). These results indicate that Mst2 and Epe1 act synergistically to restrain the H3K14ub-H3K9me3 feedback loop.

We next hypothesize that compromising the H3K14ub-H3K9me3 feedback loop would prevent activation of the adaptive response. To test this, we combined *mst2Δ* and *epe1Δ* cells with mutations that weaken the feedback loop (*clr4-W31G*, *clr4-3FA*, *hht3-K9R*, and *hht1-K14R*). In all cases, the resulting triple mutants no longer produce small colonies (Fig. 6e) and fail to induce adaptive accumulation of H3K9me2 at *clr4+* (Fig. 6f). Together, these findings demonstrate that Mst2 and Epe1 function as independent brakes that restrain the H3K14ub-H3K9me3 feedback loop and prevent uncontrolled heterochromatin expansion.

## Discussion

Our study identifies a chromatin-based feedback loop between H3K14ub and H3K9me3 that underpins heterochromatin spreading and epigenetic inheritance. We show that H3K14ub directly stimulates Clr4 methyltransferase activity on nucleosomal substrates, overcoming the inhibitory effect of nucleosomes. In turn, H3K14ub and H3K9me3 cooperate to stabilize the association of CLRC with chromatin, reinforcing deposition of both modifications. This feedback loop is tuned by Clr3-mediated H3K14 deacetylation and restrained by

opposing activities of Mst2 and Epe1. Together, these factors constitute an integrated circuit that coordinates histone ubiquitination, deacetylation, and methylation to ensure robust and heritable heterochromatin domains.

### A nucleosome-specific role for H3K14ub in stimulating Clr4 activity

While previous studies have shown that H3K14ub enhances Clr4 enzymatic activity in vitro using histone peptides[30–33], our work extends these findings to a more physiologically relevant context by showing that H3K14ub also stimulates Clr4 activity on nucleosomes (Fig. 1c). Moreover, in the absence of H3K14ub, nucleosomal DNA markedly inhibits Clr4 activity, consistent with prior observations that nucleic acids inhibit the enzymatic activity of Clr4[60].

To date, structural insights have been limited to the Clr4-H3K14ub peptide complex[30]. The molecular basis of the DNA-mediated inhibition of Clr4 and the mechanism by which H3K14ub relieves it remain unclear. A reasonable expectation is that H3K14ub induces conformational changes within the nucleosome that enhance Clr4 accessibility or positioning. Importantly, this stimulatory effect appears specific to H3K14ub and is not recapitulated by other ubiquitinated histones. These results suggest that Clr4 recognizes a precise epigenetic configuration that licenses H3K9 methylation within chromatin.

### A Self-reinforcing feedback loop between H3K14ub and H3K9me3

A central finding of our study is that H3K14ub and H3K9me3 are mutually dependent in vivo and co-localize across heterochromatic domains. Our data support a two-tiered model: H3K14ub enhances Clr4 activity to propagate H3K9me3, and both H3K14ub and H3K9me3, in turn, stabilize CLRC on chromatin. Such a self-reinforcing loop amplifies local signals and supports both lateral propagation and epigenetic inheritance of heterochromatin.

Positive feedback loops are hallmarks of bistable biological systems, enabling sharp, switching-like transitions between "on" and "off" states[9]. The reciprocal reinforcement between H3K14ub and H3K9me3 provides exactly such a mechanism: once a locus acquires either modification, the loop rapidly amplifies it, stabilizing the heterochromatin state. Conversely, loci that fail to cross this threshold remain euchromatic, thereby sharpening domain boundaries and reducing transcriptional noise.

A distinguishing feature of the H3K14ub-H3K9me3 feedback loop is its functional modulation by histone acetylation. The mutually exclusive nature of H3K14ub and H3K14ac provides a mechanism by which the acetylation landscape can shape heterochromatin formation. Clr3 deacetylase activity promotes H3K14ub by clearing acetyl marks that would otherwise block ubiquitylation. At nucleation sites, increased recruitment of CLRC buffers the effects of Clr3 loss, resulting in only a modest reduction of H3K14ub and consequently H3K9me3 levels. However, the pronounced loss of these heterochromatin marks and silencing at distal sites in clr3Δ cells underscores the importance of Clr3 in enabling the positive feedback loop to propagate heterochromatin.

Counteracting this reinforcing loop are heterochromatin antagonists such as Epe1 and Mst2, which serve to prevent ectopic or excessive heterochromatin spreading[51,53]. Epe1 promotes H3K9 demethylation[54–56] and also recruits the SAGA histone acetyltransferase complex, which acetylates H3K14[61]. Mst2 also acetylates H3K14[52], directly opposing H3K14ub deposition. Together, these factors establish a dynamic threshold: only chromatin regions that overcome these erasure mechanisms and acquire sufficient H3K14ub can initiate and maintain H3K9 methylation. This integration of positive and negative regulatory inputs defines a narrow window in which heterochromatin can form, providing a mechanistic explanation for how cells achieve spatially restricted, bistable heterochromatin domains.

Our genetic analyses further show that even partial reduction of available sites for H3K14ub or H3K9me3, via single-copy H3K14R or H3K9R mutations, is sufficient to impair the H3K14ub-H3K9me3 feedback loop and heterochromatin spreading. This suggests that local saturation of these marks is necessary to sustain read-write coupling, and that interference at even one of the multiple H3 gene copies can perturb this balance, consistent with previous studies[45,46].

### Comparison with other histone ubiquitination pathways

The stimulation of histone methylation by histone ubiquitination represents a general theme in chromatin regulation, linking distinct chromatin-modifying activities into integrated signaling circuits[62]. One of the most extensively characterized examples of ubiquitin-methylation crosstalk is H2BK120ub, which stimulates H3K4me by Set1/COMPASS and H3K79me by Dot1[63–66]. However, unlike the H3K14ub-H3K9me3 feedback, there is no reciprocal enhancement of H2Bub by histone methylation, and H2Bub-mediated methylation occurs in the context of active transcription and euchromatin, where it reinforces transcriptional elongation and open chromatin structure.

Another related circuit involves H2AK119ub and H3K27me3, catalyzed by Polycomb repressive complex 1 (PRC1)[67,68] and PRC2[69–72], respectively. In this case, mono-ubiquitination of H2A on lysine 119 promotes recruitment or stabilization of PRC2 at chromatin, facilitating deposition of H3K27me3[73,74]. Conversely, PRC2-mediated H3K27me3 enhances PRC1 recruitment[75,76], creating a self-reinforcing feedback loop that drives gene silencing in facultative heterochromatin. Like the H3K14ub–H3K9me3 feedback, this system relies on reciprocal histone modifications to sustain a repressive chromatin state.

Together, these examples illustrate how histone mono-ubiquitination can be co-opted for diverse functions across chromatin landscapes: either as a catalytic enhancer (H3K14ub, H2Bub1), a recruitment signal (H2AK119ub), or part of a self-reinforcing feedback loop (H3K14ub-H3K9me3, H2AK119ub-H3K27me3). The H3K14ub-H3K9me3 axis represents a unique convergence of these principles, integrating post-translational crosstalk, enzyme activation, and epigenetic inheritance into a single regulatory module.

### Functional implications and broader significance

Our work sheds light on how chromatin-modifying enzymes integrate distinct histone modifications into functional circuits that ensure the fidelity of epigenetic inheritance. The H3K14ub-H3K9me3 feedback loop adds an important dimension to the classic view of heterochromatin regulation, placing histone ubiquitylation on par with methylation and acetylation in establishing chromatin state.

By uncovering the H3K14ub-H3K9me3 circuit, our work provides a conceptual resolution to the long-standing puzzle of how the relatively weak H3K9me3 read-write cycle alone could sustain epigenetic inheritance. Instead, we propose that heterochromatin relies on a multi-layered network in which ubiquitination, deacetylation, and methylation are dynamically coupled to ensure both stability and flexibility. Given that both H3K14ub and H3K9me3 are conserved in mammals[21,40], this mechanism may represent a general principle by which chromatin feedback circuits enforce the bistability of repressive domains across eukaryotes.

## Methods

### Fission yeast strains

Yeast strains overexpressing Flag-raf1+ were generated by using Flag-raf1+ to replace the ade6+ coding region through CRISPR-mediated recombination[77], allowing it to be driven by the ade6 promoter. All other strains were generated by genetic crosses. A list of fission yeast strains used in this study is provided in Supplementary Data 1.

### Protein expression and purification

The SET domain of recombinant Clr4 (residues 190–490) and the 3FA mutant were cloned into a pET28a vector. The expression plasmids

were transformed into Rosetta cells, and protein expression was induced using 0.15 mM isopropyl-1-thio-D-galactopyranoside. After incubation overnight at 16 °C, the cells were harvested and resuspended in lysis buffer (50 mM sodium phosphate, 300 mM NaCl, pH 7.0), supplemented with 2 mM β-mercaptoethanol and lysed with ultrasonication. The lysate was incubated with Talon metal affinity resin (Takara Bio USA, 635501) and then washed with lysis buffer containing 15 mM imidazole. Bound protein was eluted with lysis buffer containing 150 mM imidazole.

### In vitro histone methyltransferase assays

Histone methyltransferase assays were performed with recombinant Clr4 SET domain, histone H3/H4 tetramer or nucleosomes in histone methyltransferase buffer (10 mM Tris, pH 8.0, 1 mM EDTA, 1 mM DTT, 2 µM SAM) containing $^3$H-SAM (400 nM) for 30 min at 25 °C. Wild type (16-0006), H3K14ub (16-0398), H3K18ub (16-0401), and H2BK120ub (16-0396) nucleosomes were purchased from Epicypher. The samples were resolved by SDS-PAGE and subjected to Coomassie staining to visualize the proteins and then treated with EN3HANCE (Perkin Elmer) to visualize labeled substrates.

### Serial dilution analyses

For serial dilution plating assays, ten-fold dilutions of a mid-log-phase culture were plated on the indicated medium and grown for 3 days at 30 °C.

### Chromatin immunoprecipitation (ChIP)

Log phase yeast cultures were crosslinked with 1% formaldehyde for 20 min at room temperature, followed by the addition of 125 mM glycine for 5 min. Cells were then harvested and washed with cold PBS and resuspended in ChIP lysis buffer (50 mM HEPES-KOH, pH 7.5, 140 mM NaCl, 1% Triton X-100, 0.1% Deoxycholate, 1 mM PMSF). Cold glass beads were added, and the mixture was vigorously shaken in a MiniBeadBeater (Biospec Products). The lysates were collected and sonicated with Bioruptor® Pico (Diagenode) for 12 cycles (30 s on/30 s off). After centrifugation at 16,000 g for 15 min to clarify the lysates, the released chromatin was immunoprecipitated overnight at 4 °C with antibodies: H3K14ub, H3K9me3 (Active Motif 39161), and myc (Sigma C3956). Protein G Agarose beads (Sigma 11243233001) were added for an additional 2 h at 4 °C. The beads were washed twice with ChIP lysis buffer, once each with ChIP lysis buffer containing 0.5 M NaCl, Wash buffer (10 mM Tris, pH 8.0, 250 mM LiCl, 0.5% NP-40, 0.5% Deoxycholate, 1 mM EDTA), and TE buffer (50 mM Tris, pH 8.0, 1 mM EDTA). Beads-bounded chromatin was eluted with TES buffer (50 mM Tris pH 8.0, 1 mM EDTA, 1% SDS) at 65 °C and then incubated overnight at 65 °C to reverse crosslinking. The DNA-protein mixtures were treated with Proteinase K (Invitrogen 10005393), and DNA was purified by phenol: chloroform extraction, followed by ethanol precipitation.

Quantitative PCR (qPCR) was performed using Luna Universal qPCR Master Mix (NEB M3003S) on a StepOne Plus Real-Time PCR System (Applied Biosystems). DNA serial dilutions were used as templates to generate a standard curve of amplification for each pair of primers, and the relative concentration of the target sequence was calculated accordingly. The value from wild-type cells was arbitrarily set to 1 and served as a reference for other samples. A list of DNA oligos used is provided in Supplementary Data 2.

### ChIP-seq

Sequencing reads were trimmed using Trimmomatic (v0.38) to remove Illumina adapters and low-quality sequences. Filtered reads were aligned to the *Schizosaccharomyces pombe* reference genome (ASM294v3) using BWA (v0.7.17). Alignment files were processed with SAMtools (v1.9) to filter mapped reads. Picard (v2.19.0) was used for sorting, PCR duplicate removal, and indexing. Local realignment around indels was performed with GATK3 (v3.6–6). Peak calling was performed with MACS2 (v2.2.7.1). Genome coverage was normalized to counts per million mapped reads (CPM) using bedtools (v2.27.1) to generate standardized BedGraph files. The BedGraph files were visualized using IGV. ChIP-seq data is available in the NCBI database under accession number PRJNA1300844.

### Western blot analysis

Cell lysates were prepared by lysing yeast cells with a beadbeater in ChIP lysis buffer. The cleared lysates were mixed with 2 x SDS loading dye and resolved by SDS-PAGE. Western blot analyses were performed with H3K14ub and Flag (Sigma F7425 and Genscript A00187) antibodies.

### Protein purification and co-immunoprecipitation

Log phase yeast cells were harvested and washed with 2xHC buffer (300 mM HEPES-KOH at pH 7.6, 2 mM EDTA, 100 mM KCl, 20% glycerol, 2 mM DTT, 1 mM PMSF) and frozen in liquid nitrogen. Crude cell extracts were prepared by vigorously blending frozen yeast cells with dry ice using a household blender, followed by incubation with 1xHC buffer containing 250 mM KCl for 30 min. The lysate was cleared by centrifugation at 15,000 g for 1 h. The supernatants were incubated with Flag-agarose (Sigma A2220) overnight, and washed four times with 1xHC containing 250 mM KCl. For co-immunoprecipitation analysis, bound proteins were resolved by SDS-PAGE followed by Western blot analyses with Myc (Sigma C3956) or Flag (Sigma F7425) antibodies.

### Quantification and statistical analysis

For ChIP-qPCR, data are presented as mean ± sd of three biological replicates. Statistical significance was assessed using two-tailed unpaired Student's $t$-tests for pairwise comparisons.

### Declaration of generative AI and AI-assisted technologies in the writing process

During the preparation of this work, the authors used ChatGPT to perform statistical analysis and improve language and readability. After using this tool/service, the authors reviewed and edited the content as needed and take full responsibility for the content of the publication.

### Reporting summary

Further information on research design is available in the Nature Portfolio Reporting Summary linked to this article.

## Data availability

The ChIP-seq data generated in this study have been deposited in the NCBI database under accession code PRJNA1300844. Source data are provided with this paper.

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

## Acknowledgments

We thank Ke Zhang Reid and Karl Ekwall for yeast strains and members of the Jia laboratory for helpful discussions. This work was supported by NIH grant R35-GM126910 to SJ.

## Author contributions

S.J. conceived the project and designed experiments. S.J., T.T., J.Z., and Y.F. performed experiments. Q.H. generated the H3K14ub antibody. P.J. analyzed sequencing data. S.J., J.W., and C.M.S. supervised research. S.J. wrote the manuscript with input from all authors.

## Competing interests

The authors declare no competing interests.
