## [Transparent Peer Review file · Nature Communications]

An H3K14ub-H3K9me3 feedback circuit governs heterochromatin spreading and inheritance in fission yeast

Corresponding Author: Professor Songtao Jia

Version 0:

Reviewer comments:

Reviewer #1

(Remarks to the Author)

In this manuscript, Toda et al. propose a novel mechanism to explain the long-standing question of how the vulnerable H3K9me3 read-write cycle alone can maintain epigenetic inheritance. The authors identify a H3K14ub–H3K9me3 circuit that forms a mutually reinforcing feedback loop. This work builds upon previous findings that H3K14ub is required for H3K9me3 (PMIDs: 40446033, 34524082, 31468675, 34010645).

The authors provide convincing results to support several new findings:

- 1) H3K9me3 is also necessary for H3K14ub.
- 2) Both H3K14ub and H3K9me3 stimulate Ctr4 activity to promote H3K9me3, and also contribute to recruiting and stabilizing CLRC at heterochromatin.
- 3) Any partial disruption of this circuit causes heterochromatin defects.
- 4) Impairment of the activities that antagonize H3K14ub or H3K9me3 alleviates the defects, supporting that both positive and negative feedback activities synergistically regulate the propagation and maintenance of heterochromatin.

Additionally, this study is the first to map the genome-wide distribution and biological function of H3K14ub in fission yeast. This work is original and significant, with data that support the conclusions, making it of interest to a broad audience in the field of heterochromatin.

Minor comments for the authors to consider:

Figure 1c: Please clarify what the X and Y axes represent.

Figure 1c Legend: Add "tel1L," "cen1," and "tel2R" to the legend.

Mating Type Locus: Does H3K14ub colocalize with H3K9me3 at the mating type locus, one of the three major heterochromatic regions? This locus was not shown in Figure 1c, although it was used for further analysis in Figures 3-5.

Line 40: Rea et al. (#12) should be cited with references 6–8, not 9–12.

Line 50 and 237: "Histone deacetylation, ubiquitination, and methylation" might be a more appropriate description than "histone ubiquitination, acetylation, and methylation."

Line 103–104: Data for H3K14R and deletion of raf1 and raf2 are not shown in the main or supplemental figures.

Line 132: What evidence supports the notion that H3K14ub makes a stronger contribution? The results shown in Figure 2f–2g appear to support this.

The methods section includes a quantification and statistical analysis section; however, no statistical analysis is presented in the ChIP-qPCR figures. It is also unclear what statistical methods the authors would use for comparison.

Figure 3a: Should the hht3-K9R data look different from hht1-K14R (symbol + a bar)? Does the blue bar indicate the Flag tag?

Figure 3b: Should the mutants on the H3 tail be more visible in the hht mutant?

Reviewer #2

(Remarks to the Author)

Methylation of H3K9 (H3K9me), which is catalyzed by the histone methyltransferase Clr4/Suv39, is a conserved mark that assembles a higher-order chromatin structure called heterochromatin. In fission yeast, Clr4 forms a complex called CLRC that provides H3K14 ubiquitylation (H3K14ub). Previous studies have shown that H3K14ub promotes Clr4's H3K9 methyltransferase activity *in vitro*. In this manuscript, the authors generated a specific antibody against H3K14ub. Using this antibody, the authors analyzed the genome-wide localization of H3K14ub and demonstrated that it clearly overlaps with H3K9me3. The authors also examined the relationship between the factors involved in H3K9me and H3K14ub, showing that H3K9me and H3K14ub form a feedback loop. Finally, the authors demonstrated that this feedback loop is negatively regulated by the histone acetyltransferase Mst2 and the H3K9 demethylase Epe1.

Overall comments:

Although accumulating evidence indicates that H3K14ub plays a pivotal role in depositing H3K9me in heterochromatic regions, its genome-wide localization and dynamics are not fully understood. This study used an H3K14ub antibody and provides strong evidence that H3K14ub is closely associated with H3K9me. In addition, it is important to note that this feedback loop is negatively regulated by counteracting histone-modifying enzymes. The experiments were carefully designed, and most of the conclusions are supported by the presented data. Having said that, the novelty of the findings appears limited, as most of the conclusions are predictable from previous genetic studies, and the authors merely verified them by using the H3K14ub antibody. Below, I raise several points that the authors should address to improve the manuscript.

Major points:

1. (Figs. 1 and 2):

While the authors demonstrated that Rik1 (CLRC) recruitment depends on H3K9me and H3K14ub, they did't show whether Clr4 recruitment depends on these modifications. To demonstrate interdependence, the authors should show that Clr4 localization is affected by H3K9R or H3K14R amino acid substitutions, or CLRC complex formation.

2. (Fig 4c, Fig S3c):

The finding that Rik1 levels are elevated at cenH in *clr3Δ* strains is intriguing. While the authors suggest that CLRC accumulates at cenH because it cannot spread, does Clr4 localization also increase similarly at cenH? Furthermore, cenH functions as an H3K9me initiation site by recruiting the RITS complex (doi: 10.1038/ng1452). It would be helpful to discuss whether the RNAi pathway is involved in the increased Rik1 (CLRC) localization at cenH observed in *clr3Δ* cells.

3. (Fig. 5):

The authors noted that negative regulators, such as Mst2 and Epe1, restrict the H3K14ub-H3K9me3 feedback loop, thereby preventing the uncontrolled expansion of heterochromatin. However, they demonstrated heterochromatin expansion only at marker genes inserted in heterochromatic regions. To clarify the roles of Mst2 and Epe1 against the H3K14ub-H3K9me3 feedback loop under physiological conditions, the authors could perform ChIP-seq analyses to test whether H3K14ub and H3K9me3 expand beyond well-defined heterochromatic regions in *mst2Δ* and *epe1Δ* mutants, as shown in Fig. 1c.

4. (lines 48 and 204):

The authors describe the actions of Mst2 and Epe1 as negative feedback. While these factors negatively regulate the H3K9me-H3K14ub feedback loop, this is not sufficient to describe it as negative feedback. To do so, the authors need to demonstrate a relationship in which H3K14 acetylation by Mst2 promotes H3K9 demethylation, and, conversely, H3K9 demethylation promotes H3K14 acetylation by Mst2.

Minor points:

1. (Abstract, lines 39-40):

References #11 and #12 do not include any results regarding H3K14 ubiquitylation activity of CLRC.

2. (Fig. 1c):

Although the authors clearly showed the genome-wide localization of H3K14ub by ChIP-seq analysis using H3K14ub antibody, its heterochromatic localization and coexistence with H3K9me have been previously demonstrated by ChIP-MS/MS analysis (doi: 10.15252/embr.201948111). The authors should refer to the published study in the text.

3. (Fig. 1c):

H3K14ub and H3K9me3 are not detected at the mat region of chromosome 2. If this is the case, the author should accurately describe it and provide possible reasons, such as the strain used.

4. (ChIP-qPCR):

To determine significant or non-significant differences, it would be better to apply statistical analyses to ChIP-qPCR data.

5. (Fig. S2):

Ubiquitylated nucleosomes were used as the substrate for *in vitro* histone methyltransferase assays. However, the Methods section does not include the description how the ubiquitylated nucleosomes were reconstituted.

6. (Reference #34):
The authors should provide the correct citation.

Reviewer #3

(Remarks to the Author)

Toda et al. present an analysis of the role of the histone modification H3K14ub in heterochromatin formation in fission yeast. This is a very neat genetic analysis and reveals that H3K14ub and H3K9me_{2/3} depend on each other in a finely balanced system that involves the H3K14ac and H3K9me.

This analysis brings long-awaited clarity about the role of H3K14ub in the heterochromatin and gene silencing system of fission yeast. Due to its highly conserved machinery, these results are likely to be relevant for many other systems, including human cells, where the significance of H3K14ub in heterochromatin has now been well established. The characterization of the feedback loop using highly targeted mutants reveals novel dependencies, showing that H3K9me is critical for H3K14ub and vice versa.

The data is of high quality and clarity and is presented in a consistent and clear manuscript. Provided the concerns below are addressed, this manuscript is highly adequate for publication in Nature Communications.

Issues:

- Statistics for the ChIP-qPCR data are specified as mean \pm SD of 2 technical replicates. This is inadequate and should be based on at least 3 biological replicates.
- The antibody is custom-made and there is no information on how the authors are planning to share it. Given China's strict export rules, there should be a provision for distribution and some information on how the antibody can be obtained by the international research community.

Version 1:

Reviewer comments:

Reviewer #1

(Remarks to the Author)

The revised version of the manuscript has completely addressed my concerns. I have no further suggestions for the authors.

Reviewer #2

(Remarks to the Author)

The authors thoroughly addressed all of my concerns, and their rebuttal statements convinced me of the novelty of this work. I therefore recommend publishing this article in its current form in Nature Communications.

Reviewer #3

(Remarks to the Author)

My concerns have been addressed.

We thank the reviewers for their time and constructive comments on our manuscript. In response, we have performed additional experiments and revised the text to address their concerns. The manuscript has also been reformatted to conform to the style of *Nature Communications*, with expanded Introduction and Discussion sections, and the conversion of Figure S1 to Figure 1. Below, we provide a detailed, point-by-point response to the reviewers' comments.

Reviewer #1 (Remarks to the Author):

In this manuscript, Toda et al. propose a novel mechanism to explain the long-standing question of how the vulnerable H3K9me3 read-write cycle alone can maintain epigenetic inheritance. The authors identify a H3K14ub–H3K9me3 circuit that forms a mutually reinforcing feedback loop. This work builds upon previous findings that H3K14ub is required for H3K9me3 (PMIDs: 40446033, 34524082, 31468675, 34010645).

The authors provide convincing results to support several new findings:

- 1) H3K9me3 is also necessary for H3K14ub.
- 2) Both H3K14ub and H3K9me3 stimulate Clr4 activity to promote H3K9me3, and also contribute to recruiting and stabilizing CLRC at heterochromatin.
- 3) Any partial disruption of this circuit causes heterochromatin defects.
- 4) Impairment of the activities that antagonize H3K14Ub or H3K9me3 alleviates the defects, supporting that both positive and negative feedback activities synergistically regulate the propagation and maintenance of heterochromatin.

Additionally, this study is the first to map the genome-wide distribution and biological function of H3K14ub in fission yeast. This work is original and significant, with data that support the conclusions, making it of interest to a broad audience in the field of heterochromatin.

We thank the reviewer for this insightful and highly positive assessment of our work. We appreciate their recognition of the novelty, significance, and broad relevance of our findings.

Minor comments for the authors to consider:

Figure 1c: Please clarify what the X and Y axes represent.

We have clarified the identities of X (genome coordinates) and Y (reads per million mapped reads) axes in the revised legend (now Figure 2c).

Figure 1c Legend: Add "tel1L," "cen1," and "tel2R" to the legend.

We have added description of "tel1L" and "cen1", and "mat" (which replaces tel2R) in the revised legend (now Figure 2c).

Mating Type Locus: Does H3K14ub colocalize with H3K9me3 at the mating type locus, one of the three major heterochromatic regions? This locus was not shown in Figure 1c, although it was used for further analysis in Figures 3-5.

The silent mating-type was not included in the genome assembly because commonly used laboratory strains often contain rearrangements at this locus. The strain used in our study carries the *mat1-Msmt0* configuration, which retains an intact silent *mat* region. We have now incorporated this region into the analysis and find that both H3K14ub and H3K9me3 are detected at this locus (new Figure 2c).

Line 40: Rea et al. (#12) should be cited with references 6–8, not 9–12.

References have been removed from the abstract in the revised manuscript. Therefore, this comment no longer applies.

Line 50 and 237: "Histone deacetylation, ubiquitination, and methylation" might be a more appropriate description than "histone ubiquitination, acetylation, and methylation."

We changed the text accordingly to reflect the reviewer's suggestion.

Line 103–104: Data for H3K14R and deletion of *raf1* and *raf2* are not shown in the main or supplemental figures.

We now explicitly refer to Fig. 2d, e, where the data for *H3K14R* and the *raf1Δ* and *raf2Δ* mutants are shown.

Line 132: What evidence supports the notion that H3K14ub makes a stronger contribution? The results shown in Figure 2f–2g appear to support this.

Because statistical analysis did not reveal a significant difference between the *clr4-3FA* and *clr4-W31G* mutants, we have removed this statement from the revised text.

The methods section includes a quantification and statistical analysis section; however, no statistical analysis is presented in the ChIP-qPCR figures. It is also unclear what statistical methods the authors would use for comparison.

We have now performed three biological replicates and statistical analyses for the ChIP-qPCR data and assessed significance using two-tailed unpaired Student's *t*-tests. The statistical annotations have been added to corresponding figures.

Figure 3a: Should the hht3-K9R data look different from hht1-K14R (symbol + a bar)? Does the blue bar indicate the Flag tag?

In the revised Figure 4a, we used red star to indicate *K14R*, green star to indicate *K9R*, and blue bar to indicate the Flag tag. We have updated the figure legend to clarify these details.

Figure 3b: Should the mutants on the H3 tail be more visible in the hht mutant?

We have adjusted the figure to make the H3 tail mutants more clearly visible.

Reviewer #2 (Remarks to the Author):

Methylation of H3K9 (H3K9me), which is catalyzed by the histone methyltransferase Clr4/Suv39, is a conserved mark that assembles a higher-order chromatin structure called heterochromatin. In fission yeast, Clr4 forms a complex called CLRC that provides H3K14 ubiquitylation (H3K14ub). Previous studies have shown that H3K14ub promotes Clr4's H3K9 methyltransferase activity in vitro. In this manuscript, the authors generated a specific antibody against H3K14ub. Using this antibody, the authors analyzed the genome-wide localization of H3K14ub and demonstrated that it clearly overlaps with H3K9me3. The authors also examined the relationship between the factors involved in H3K9me and H3K14ub, showing that H3K9me and H3K14ub form a feedback loop. Finally, the authors demonstrated that this feedback loop is negatively regulated by the histone acetyltransferase Mst2 and the H3K9 demethylase Epe1.

Overall comments:

Although accumulating evidence indicates that H3K14ub plays a pivotal role in depositing H3K9me in heterochromatic regions, its genome-wide localization and dynamics are not fully understood. This study used an H3K14ub antibody and provides strong evidence that H3K14ub is closely associated with H3K9me. In addition, it is important to note that this feedback loop is negatively regulated by counteracting histone-modifying enzymes. The experiments were carefully designed, and most of the conclusions are supported by the presented data. Having said that, the novelty of the findings appears limited, as most of the conclusions are predictable from previous genetic studies, and the authors merely verified them by using the H3K14ub antibody. Below, I raise several points that the authors should address to improve the manuscript.

The reviewer notes that “the experiments were carefully designed and most of the conclusions are supported by the presented data,” and we thank the reviewer for this positive assessment. We respectfully disagree, however, with the concern that “the novelty of the findings appears limited.” While prior genetic studies suggested that H3K14ub depends on the Cul4-Rik1-Raf1-Raf2 E3 ubiquitin ligase, our study reveals an unexpected and previously unrecognized requirement for Clr4 enzymatic activity in establishing H3K14ub. This finding could not have been inferred from existing genetic data and directly supports our central conclusion that H3K9me3 and H3K14ub form a self-reinforcing feedback loop. Importantly, the use of an H3K14ub-specific antibody enables direct genome-wide and locus-specific interrogation of this modification, moving beyond genetic inference and previous mass spectrometry studies. This allows us to define the spatial coupling and regulatory logic of H3K14ub and H3K9me3 in vivo, providing mechanistic insight into how heterochromatin integrity is maintained. We therefore believe that the work offers a substantive advance rather than a simple confirmation of prior genetic models.

Major points:

1. (Figs. 1 and 2):

While the authors demonstrated that Rik1 (CLRC) recruitment depends on H3K9me and H3K14ub, they didn't show whether Clr4 recruitment depends on these modifications. To demonstrate interdependence, the authors should show that Clr4 localization is affected by H3K9R or H3K14R amino acid substitutions, or CLRC complex formation.

We performed ChIP analysis of Clr4, and the new data (new Figure S2a) show that Clr4 localization at *dh* repeats is abolished by both *H3K9R* and *H3K14R* mutations. To assess whether these mutants affect CLRC assembly, we also performed co-immunoprecipitation analyses of Clr4 and Rik1 and found that their interaction is not affected by either the *H3K9R* or *H3K14R* mutations (new Figure S2b).

2. (Fig 4c, Fig S3c):

The finding that Rik1 levels are elevated at *cenH* in *clr3Δ* strains is intriguing. While the authors suggest that CLRC accumulates at *cenH* because it cannot spread, does Clr4 localization also increase similarly at *cenH*? Furthermore, *cenH* functions as an H3K9me initiation site by recruiting the RITS complex (doi: 10.1038/ng1452). It would be helpful to discuss whether the RNAi pathway is involved in the increased Rik1 (CLRC) localization at *cenH* observed in *clr3Δ* cells.

We performed ChIP analysis of Clr4, and the new data (new Figure S3d-f) show that Clr4 localization at *dh* and *cenH* is increased in *clr3Δ* cells, similar to what we observe for Rik1. As the reviewer points out, the Rik1 at *cenH* in *clr3Δ* cells may result from increased transcription at this locus. At present, we cannot distinguish whether CLRC accumulation arises primarily from impaired spreading away from initiation sites or from increased transcription-dependent recruitment via the RNAi pathway. Importantly, however, this ambiguity does not affect our conclusion that CLRC fails to spread to distal regions at the mating-type locus in *clr3Δ* cells. We have now cited the referred study in the revised manuscript.

3. (Fig. 5):

The authors noted that negative regulators, such as Mst2 and Epe1, restrict the H3K14ub-H3K9me3 feedback loop, thereby preventing the uncontrolled expansion of heterochromatin. However, they demonstrated heterochromatin expansion only at marker genes inserted in heterochromatic regions. To clarify the roles of Mst2 and Epe1 against the H3K14ub-H3K9me3 feedback loop under physiological conditions, the authors could perform ChIP-seq analyses to test whether H3K14ub and H3K9me3 expand beyond well-defined heterochromatic regions in *mst2Δ* and *epe1Δ* mutants, as shown in Fig. 1c.

We thank the reviewer for this thoughtful suggestion regarding the assessment of H3K14ub and H3K9me3 spreading under physiological conditions. We have previously performed genome-wide analyses of H3K9 methylation in *mst2Δ* and *epe1Δ* single-mutant cells, which showed only modest effects on heterochromatin spreading (Wang *et al.*, 2015). In contrast, robust and readily detectable heterochromatin expansion was observed in *mst2Δ epe1Δ swi6Δ* cells. The *mst2Δ epe1Δ swi6Δ* background was used in place of a clean *mst2Δ epe1Δ* background because *mst2Δ epe1Δ* cells rapidly acquire a compensatory epigenetic suppressor that silences *clr4+*, allowing cells to revert toward a wild-type like chromatin state (Wang *et al.*, 2015). Deletion of *swi6+* prevents this adaptive response and permits stable analysis of heterochromatin expansion.

Rather than repeating genome-wide profiling, we therefore focused on targeted ChIP-qPCR at representative loci that sensitively report heterochromatin expansion: a site outside the *IRC1R* boundary and the facultative heterochromatin island *mei4+*. We show that combined loss of Mst2 and Epe1 leads to marked expansion of both H3K14ub and H3K9me3 in *mst2Δ epe1Δ swi6Δ* cells (new Figure S4a,b), supporting the conclusion that Mst2 and Epe1 cooperate to restrain the H3K14ub-H3K9me3 regulatory circuit.

4. (lines 48 and 204):

The authors describe the actions of Mst2 and Epe1 as negative feedback. While these factors negatively regulate the H3K9me-H3K14ub feedback loop, this is not sufficient to describe it as negative feedback. To do so, the authors need to demonstrate a relationship in which H3K14 acetylation by Mst2 promotes H3K9 demethylation, and, conversely, H3K9 demethylation promotes H3K14 acetylation by Mst2.

We thank the reviewer for this comment. We agree that while our data support a role for Mst2 and Epe1 as negative regulators of heterochromatin, they do not meet the strict criteria for a negative feedback loop. This might be confused with Mst2 and Epe1 together elicit the formation of ectopic heterochromatin at *clr4+*, which forms negative feedback. To avoid any confusion, we have therefore revised the manuscript to describe the actions of Mst2 and

Epe1 as antagonistic or counteracting activities that negatively regulate heterochromatin spreading and maintenance, without any referencing negative feedback loop.

Minor points:

1. (Abstract, lines 39-40):

References #11 and #12 do not include any results regarding H3K14ub ubiquitylation activity of CLRC.

References have been removed from the abstract in the revised manuscript. Therefore, this comment no longer applies.

2. (Fig. 1c):

Although the authors clearly showed the genome-wide localization of H3K14ub by ChIP-seq analysis using H3K14ub antibody, its heterochromatic localization and coexistence with H3K9me have been previously demonstrated by ChIP-MS/MS analysis (doi: 10.15252/embr.201948111). The authors should refer to the published study in the text.

We thank the review for pointing this out. We agree with the reviewer and have now cited this study (doi: 10.15252/embr.201948111) in the section describing our ChIP-seq analysis.

3. (Fig. 1c):

H3K14ub and H3K9me3 are not detected at the *mat* region of chromosome 2. If this is the case, the author should accurately describe it and provide possible reasons, such as the strain used.

The silent mating-type was not included in the genome assembly because commonly used laboratory strains often contain rearrangements at this locus. The strain used in our study carries the *mat1-Msmt0* configuration, which retains an intact silent *mat* region. We have now incorporated this region into the analysis and find that both H3K14ub and H3K9me3 are detected at this locus (new Figure 2c).

4. (ChIP-qPCR):

To determine significant or non-significant differences, it would be better to apply statistical analyses to ChIP-qPCR data.

We have now performed three biological replicates and statistical analyses for the ChIP-qPCR data and assessed significance using two-tailed unpaired Student's *t*-tests. The statistical annotations have been added to corresponding figures.

5. (Fig. S2):

Ubiquitylated nucleosomes were used as the substrate for in vitro histone methyltransferase assays. However, the Methods section does not include the description how the ubiquitylated nucleosomes were reconstituted.

The ubiquitylated nucleosomes were commercially obtained from Epicypher. We have now added this information to the method section of the revised manuscript.

6. (Reference #34):

The authors should provide the correct citation.

We have updated the citation to reference 34, which is now published.

Reviewer #3 (Remarks to the Author):

Toda et al. present an analysis of the role of the histone modification H3K14ub in heterochromatin formation in fission yeast. This is a very neat genetic analysis and reveals that H3K14ub and H3K9me_{2/3} depend on each other in a finely balanced system that involves the H3K14ac and H3K9me.

This analysis brings long-awaited clarity about the role of H3K14ub in the heterochromatin and gene silencing system of fission yeast. Due to its highly conserved machinery, these results are likely to be relevant for many other systems, including human cells, where the significance of H3K14ub in heterochromatin has now been well established. The characterization of the feedback loop using highly targeted mutants reveals novel dependencies, showing that H3K9me is critical for H3K14ub and vice versa.

The data is of high quality and clarity and is presented in a consistent and clear manuscript. Provided the concerns

below are addressed, this manuscript is highly adequate for publication in Nature Communications.

We thank the reviewer for this thoughtful and positive evaluation of our work. We appreciate the reviewer's recognition of the clarity, quality, and significance of our genetic analysis, as well as the identification of the mutually dependent relationship between H3K14ub and H3K9me in heterochromatin formation. We are encouraged that the reviewer finds this study provides long-awaited mechanistic insight into the role of H3K14ub in fission yeast and notes its broader relevance to conserved heterochromatin systems.

Issues:

- Statistics for the ChIP-qPCR data are specified as mean +/- SD of 2 technical replicates. This is inadequate and should be based on at least 3 biological replicates.

We have now performed three biological replicates and statistical analyses for the ChIP-qPCR data and assessed significance using two-tailed unpaired Student's *t*-tests. The statistical annotations have been added to corresponding figures.

- The antibody is custom-made and there is no information on how the authors are planning to share it. Given China's strict export rules, there should be a provision for distribution and some information on how the antibody can be obtained by the international research community.

We appreciate the reviewer's concern regarding access to the H3K14ub antibody. The antibody was custom generated and is available in limited quantities. We are committed to facilitating its use by the research community to the extent possible and will make reasonable efforts to share the reagent with interested investigators upon request.